# Investigating *Cryptosporidium* spp. Using Genomic, Proteomic and Transcriptomic Techniques: Current Progress and Future Directions

**DOI:** 10.3390/ijms241612867

**Published:** 2023-08-16

**Authors:** Joanna Dąbrowska, Jacek Sroka, Tomasz Cencek

**Affiliations:** Department of Parasitology and Invasive Disease, National Veterinary Research Institute, Partyzantów 57 Avenue, 24-100 Puławy, Polandtcencek@piwet.pulawy.pl (T.C.)

**Keywords:** *Cryptosporidium* spp., genomics, proteomics

## Abstract

Cryptosporidiosis is a widespread disease caused by the parasitic protozoan *Cryptosporidium* spp., which infects various vertebrate species, including humans. Once unknown as a gastroenteritis-causing agent, *Cryptosporidium* spp. is now recognized as a pathogen causing life-threatening disease, especially in immunocompromised individuals such as AIDS patients. Advances in diagnostic methods and increased awareness have led to a significant shift in the perception of *Cryptosporidium* spp. as a pathogen. Currently, genomic and proteomic studies play a main role in understanding the molecular biology of this complex-life-cycle parasite. Genomics has enabled the identification of numerous genes involved in the parasite’s development and interaction with hosts. Proteomics has allowed for the identification of protein interactions, their function, structure, and cellular activity. The combination of these two approaches has significantly contributed to the development of new diagnostic tools, vaccines, and drugs for cryptosporidiosis. This review presents an overview of the significant achievements in *Cryptosporidium* research by utilizing genomics, proteomics, and transcriptomics approaches.

## 1. Introduction

*Cryptosporidium* spp. is a protozoan parasite capable of infecting various vertebrates, including humans, causing gastroenteritis with symptoms such as watery diarrhea, abdominal pain, and weakness [1]. Immunocompromised individuals are particularly vulnerable to this parasite, which has a complex life cycle involving asexual and sexual reproduction within a single host and transmission through direct and indirect contact—animal-to-human, water, and food [2]. Of the over 39 species within the genus *Cryptosporidium*, *C. parvum* and *C. hominis* are the most common in humans, responsible for 90% of cases worldwide. The distribution of *Cryptosporidium* species varies across different regions globally, with *C. hominis* being the main causative agent in low-income and industrialized nations and *C. parvum* being more frequently detected in humans from Middle Eastern countries. Some species, such as *C. parvum* and *C. meleagridis*, can infect both mammals and birds, while others, including *C. parvum*, *C. ubiquitum*, and *C. muris*, can cause zoonotic mixed infections [3].

*Cryptosporidium* was first identified in mice in 1907 and later emerged as a significant causative agent of severe diarrhea in AIDS patients during the 1980s and 1990s [4]. Cryptosporidiosis is an emerging infectious disease of global public health significance, with prevalence estimates ranging from 2.6% to 21.3% in Africa and from 0.3% to 4.3% in North America [5]. Diagnosis of cryptosporidiosis is primarily based on microscopic identification of parasite oocysts, oocyst antigens, or oocyst DNA in fecal samples. However, due to the lack of specific morphological differences between parasite species, molecular tests utilizing genetic markers like SSU rRNA, COWP, hsp70, and gp60 have become widely accepted as the gold standard for species or genotype identification. It is important to note that relying solely on a few genetic markers for diagnosis may be insufficient and inaccurate, particularly when identifying mixed infections with different species and genotypes [4].

Therefore, newer molecular tools, such as whole-genome sequencing (WGS), are being employed in *Cryptosporidium* research [6]. WGS offers multiple benefits, including genetic variant identification, valuable drug response data, and the identification of therapeutic targets. It also provides insights into population genetics and evolution through the analysis of genetic variations and diversity within and between populations. Moreover, WGS aids in identifying resistance genes, understanding pathogenic mechanisms, and tracking the spread of infectious diseases, making it valuable for epidemiological studies.

When facing parasitic infections, understanding the proteins involved in the host–pathogen interaction and uncovering potential drug targets can be crucial in reducing their impact on the environment. Despite significant efforts to develop experimental models and assess nearly 1000 chemotherapeutic agents, a fully effective anti-*Cryptosporidium* therapy remains elusive [7]. The reason for the lack of drug efficacy is likely multifaceted and may include the parasite’s location within the host cell, the distinct structural and biochemical composition of drug targets, or its ability to block the import or rapidly efflux drug molecules [5]. Hence, proteomic studies play a crucial role in comprehending the fundamental mechanisms by which drugs are transported to the parasite and identifying unique targets. These steps are pivotal in the development of effective therapeutic agents.

The transcriptome refers to the complete set of RNA transcripts in a cell and is incredibly dynamic, representing all actively expressed genes at a specific moment. Analyzing the expression profiles of mRNA transcripts in a group of cells provides valuable insights into their functional processes. Gene expression profiles vary depending on the cell type, developmental stage of the organism, and parasite–host interaction, which is crucial in *Cryptosporidium* studies. In contrast to the relatively stable genome, the transcriptome is highly flexible and responsive to biological changes.

This review aims to highlight significant milestones in *Cryptosporidium* research, which we have subjectively selected, considering them as groundbreaking and contributing new insights. These milestones have been achieved through the application of genomics, proteomics, and transcriptomics techniques over the years. Our paper targets researchers, students, and anyone seeking valuable resources to attain a thorough understanding of *Cryptosporidium’s* genomics, proteomics, and transcriptomics, without the necessity of reading all individual research manuscripts separately. By presenting a comprehensive analysis of the existing literature, we strive to offer readers a more informed perspective on the subject matter.

## 2. Genomics of *Cryptosporidium* spp.

### 2.1. Exploring Cryptosporidium Diversity and Evolutionary History through Whole Genome Sequencing

#### 2.1.1. Early Studies on the *Cryptosporidium* Genome Preceding the Genomic Era

The study of *Cryptosporidium* spp. did not gain significant attention until the 1980s, resulting in a relatively late start for genome research. The first attempts at genome analysis were made in 1999 by Liu et al. [8], marking an initial step, although it did not fully elucidate the parasite’s genomic landscape. Genes were identified independently using random sequence analysis, and the assembled sequences formed 408 contigs, encompassing over 250 kb of unique sequence, representing approximately 2.5% of the *C. parvum* genomic sequence. Among these contigs, 107 (26%) showed similarity to known proteins, rRNA, and tRNA genes, including putative genes involved in the glycolytic pathway, DNA, RNA, protein metabolism, and signal transduction pathways. Additionally, they discovered diverse microsatellite sequences constituting less than 1% of the genome, which had potential as genetic markers for strain typing. The study was limited by the scarcity of gene sequences available in public databases at that time, which constrained the amount of data they could analyze. Nevertheless, despite these limitations, the study provided the first comprehensive molecular insights into the fundamental biology and cellular metabolism of this apicomplexan parasite, which has traditionally been challenging to investigate experimentally.

At the same time, Strong et al. published the first cDNA sequence survey of *C. parvum* sporozoites, generating approximately 2000 sequence tags from the Iowa isolate [6]. To expedite gene discovery and identify potential drug and vaccine targets, the authors constructed libraries using single-pass sequencing of random clones. Analysis of 567 expressed sequence tags (ESTs) and 1507 genome survey sequences (GSSs) revealed 1 megabase (1 mb) of unique genomic sequence, representing about 10% of the *C. parvum* genome. The results of the studies were crucial and provided a basis for further analysis, particularly as they were conducted before the entire genome was known. Despite this, BLAST analyses revealed that 32% of the ESTs exhibited similarity to sequences already present in publicly available databases, indicating a significant proportion of the sequences had a known homology and function. The authors’ identification of tags encoding proteins with clear therapeutic potential, such as S-adenosylhomocysteine hydrolase, histone deacetylase, polyketide/fatty-acid synthases, various cyclophilins, thrombospondin-related cysteine-rich protein, and ATP-binding-cassette transporters, was ground-breaking. These molecules, including transporters, have the potential to play a role in the rapid efflux of drugs and nutrient uptake.

#### 2.1.2. The *Cryptosporidium* Genomic Era: Opening Paths for Future Analyses

To facilitate future epidemiological studies and enhance molecular diagnostics of *Cryptosporidium* spp., WGS projects have been conducted since 2004 and continued in subsequent years. The initial projects involved extracting oocyst DNA from infected germ-free neonatal calves and utilizing the random shotgun Sanger sequencing approach. These pioneering studies marked the advent of the genomic era in *Cryptosporidium* research, laying the foundation for comprehensive future analyses. 

Two significant genomic analyses were conducted, offering valuable insights into the genomes of *C. parvum* Iowa and *C. hominis* TU502. The authors utilized the previously developed “HAPPY” mapping, a fast PCR-based in vitro methodology that allowed them to identify eight chromosomes ranging from ∼0.9 to 1.4 Mb. The *Cryptosporidium* genome was approximately 9.1 Mb in size with 13× genome coverage and five gaps. Notably, it possessed a GC content of approximately 31%, contrasting with 19.4% in *Plasmodium* and 52% in *Toxoplasma*. Approximately 5% of the 3807 predicted protein-coding genes in this dataset contain intron. The studies identified novel classes of cell-surface and secreted proteins with potential pathogenic roles, along with paralogous proteins involved in antibiotic transport as potential drug targets. Both species lack variant surface antigens used for immune evasion by other Apicomplexa and rely on a glycolysis-based metabolism but lack enzymes for synthesizing simple sugars, amino acids, and nucleotides, utilizing unique enzymes as therapeutic targets. Abrahamsen’s study revealed for the first time unique characteristics such as significantly reduced genome size and a lack of an apicoplast, which were confirmed by Xu. Additionally, the research unveiled reduced or absent genes in metabolic pathways and organelles, indicating their reliance on the host for nutrient acquisition. *C. hominis* isolate TU502 showed strong similarities in gene complement with *C. parvum*, with only 3–5% sequence divergence. A comprehensive examination of coding sequences in both species identified 11 additional protein-coding sequences in *C. hominis* not present in *C. parvum* Iowa, along with 5 unique protein-coding sequences in *C. parvum*. Among these 11 CDS, 9 are hypothetical proteins. Prioritizing the study of these proteins, particularly those from *C. hominis*, will shed light on potential transmission dynamics and host adaptation mechanisms within this genus. 

#### 2.1.3. Draft Genome Sequencing to Enhance Our Understanding of *Cryptosporidium* Parasites

Over time, as techniques for sequencing have become more sophisticated and accessible, new WGS data from additional isolates have been published. Several recent studies have utilized draft genome sequencing to improve our understanding of *Cryptosporidium* parasites.

Ifeonu et al. [7] studied different *Cryptosporidium* isolates, including *C. hominis* TU502 2012, *C. meleagridis* UKMEL1, and *C. baileyi* TAMU-09Q1. They improved the genome assembly of TU502 2012, reducing the number of contigs by 119 compared to 2004, with no change in genome size (9.1 Mb). Gene length increased by 500 bp from the previous annotation. However, the current gene set in TU502 2012 showed bias, with only 63% of the 3745 predicted proteins present in all other annotated *Cryptosporidium* genomes. Moreover, 110 predicted protein-coding genes were unique to the newly sequenced genomes and not found in the predicted proteome of *C. parvum*. The researchers identified 10,526 SNPs in TU502 2012 and 4394 in *C. hominis* UKH1. These findings underscore the need for further optimization of *Cryptosporidium* genome assemblies and additional analyses to understand the differences contributing to various phenotypes in these parasites, such as host range, infectivity, and pathogenesis.

In 2018, Nash et al. [8] sequenced the whole genome of *C. parvum* UKP1, isolated from a patient with cryptosporidiosis. The *C. parvum* IIc gp60 UKP1 type IIaA17G1R1 is a human-adapted subtype found mainly in humans and European hedgehogs. The study aimed to identify markers for transmission routes and potential virulence traits. They used 454 GS FLX Titanium and Illumina HiSeq 2500 technologies, generating 1.6 million reads, which were mapped against a reference *C. parvum* isolate (Iowa II). The parasite’s 8 chromosomes were assembled into 14 contigs, resulting in a genome size of 8,881,956 bp, with a G+C content of 30.20%, an N50 value of 1,092,230 bp, and the largest contig length of 1,333,759 bp.

#### 2.1.4. Utilizing the Latest Genomic Advancements for *Cryptosporidium* Research

Single-cell genomics is a valuable method for sequencing genomes, especially for hard-to-culture unicellular organisms like *Cryptosporidium*. Troell et al. [9] conducted an important study where they sequenced 10 single-cell genomes from clinical samples of *Cryptosporidium*-infected hosts using fluorescence-activated cell sorting (FACS) and Illumina MiSeq sequencing. The combined data achieved high coverage (99.8% at 20×) of the reference genome (*C. parvum* Iowa II). The single-cell genomes contained an average of 1.3 Gbp per sample, with minimal differences from the metagenome, indicating the accuracy of this approach for future studies. The single-cell sequencing method also enables the examination of mixed infections and the detection of multiple species/subtypes.

Another study in Bangladesh used long-read whole-genome sequencing (Pacific Biosciences) on 63 samples from infants over two years [10]. They found unexpected diversity within *C. hominis*, with numerous SNPs identified in specific regions, including the gp60 gene and sequences encoding proteins like COPS and insulinase-like peptidase. Recombination in the pathogen was observed, suggesting strong selective pressure on polymorphic regions, potentially influencing host–parasite interactions and immune evasion. To provide a clearer perspective on the sequencing and results generated in *Cryptosporidium* genomics research, the following table (Table 1) summarizes the key aspects of these studies.

### 2.2. Unraveling Cryptosporidium’s Secrets through Comparative Genomics

The genomic sequencing of *C. parvum* conducted by Abrahamsen et al. [11] revealed crucial insights into the intraspecies differences and host specificity of *Cryptosporidium*. Continuing and increasing this direction, Widmer et al. [12] employed a WGS approach that utilized single nucleotide polymorphisms (SNPs) to investigate the anthroponotic *C. parvum* isolate TU114 (subtype IIcA5G3b) in detailed comparison to the reference *C. parvum* Iowa (subtype IIaA15G2R1) and *C. hominis* TU502 (subtype IaA25R3). The analyses identified 12,000 SNPs of *C. parvum* TU114, which were significantly more similar to *C. hominis* than to *C. parvum* Iowa, despite the two *C. parvum* genome sequences being tenfold more similar to each other than either was to *C. hominis* overall. This suggests that host preference might be associated with some divergent genetic loci. Highly divergent regions were found near the ends of many chromosomes, containing genes that were almost twice as large as the genome-wide average. These genes were mainly transporters, particularly ABC transporters, and proteins with signal peptides. Furthermore, the identification of three genes (cgd1_650, cgd3_3370, and cgd6_5260) exhibiting significant allelic similarity between TU114 and *C. hominis* suggests that their evolution may be attributed to the parasite’s adaptation to distinct host species.

Guo et al. [13] studied *Cryptosporidium chipmunk* genotype I, a newly emerged zoonotic pathogen in some industrialized nations. The authors utilized WGS to investigate the transmission of the parasite from rodents to humans, and they identified the gp60 gene and nucleotide sequence encoding another mucin protein, the ortholog of cgd1_470 in *C. parvum*, by mapping to the reference genome of *C. parvum*. These molecular markers are useful for subtyping within the species due to their high sequence polymorphism. In their study, the authors characterized chipmunk genotype I in 25 human samples, 1 specimen each from an eastern gray squirrel, a chipmunk, and a deer mouse, and 4 water samples. Based on the WGS results, genotyping succeeded in generating 15 subtypes among the 30 isolates analyzed from humans, wildlife, and storm water. At the mucin locus, two subtypes were obtained: MCI and MCII, with differences in the number of a 30-bp minisatellite repeat. The authors’ findings provide valuable information for understanding the transmission dynamics of *Cryptosporidium chipmunk* genotype I and may aid in the development of effective control strategies.

*C. hominis* subtypes IbA10G2 and IaA28R4 are known to be highly virulent and have a worldwide distribution. Guo et al. [14] conducted a study in which the genomes of two field specimens of these subtypes were sequenced using both 454 and Illumina technologies and compared. The reference genome used for the analysis was *Cryptosporidium parvum* Iowa. The results revealed that 8.59–9.05 Mb of *Cryptosporidium* sequences in 45–767 assembled contigs were generated from four DNA isolates, with a coverage of 94.36–99.47%. The genomes showed almost 97% nucleotide sequence identity with the genome of *C. parvum*. Major insertions and deletions (InDels) between *C. hominis* and *C. parvum* genomes were found in the telomeric regions. Comparative analysis of the four genomes of *C. hominis* and *C. parvum* revealed differences in the 5′ and 3′ ends of chromosome 6 and the gp60 region; such differences were largely the result of genetic recombination.

In 2017, Feng et al. [15] expanded our understanding of the genetic factors involved in host adaptation by *C. parvum*, specifically those related to proteins encoded by certain genes. The genomes of three *C. parvum* isolates from China and Egypt were sequenced using the Illumina Genome Analyzer IIx. Significant genomic differences were found between the sequenced genomes and the reference Iowa genome with the 5191–5766 single nucleotide variants and the majority occurring in subtelomeric regions, including those encoding SKSR secretory proteins, the MEDLE family of secretory proteins, and insulinase-like proteases of chromosomes 1, 4, and 6. To infer the role of subtelomeric gene duplications in host adaptation by *C. parvum* subtype families, it is necessary to conduct a direct comparative genomic analysis of specimens from regions with known distinct distributions of these two subtype families among calves and lambs. Mucin proteins and other families of secretory proteins encoding invasion were found to be the most polymorphic among *C. parvum* isolates. These proteins play a role in sporozoite invasion and are highly immunogenic, making genes encoding them naturally polymorphic.

In a further comparative study of *Cryptosporidium* genomes in 2020, the authors used a comprehensive phylogenomic approach to analyze *C. hominis*, *C. parvum*, and *C. meleagridis* genomes that occur in humans [16]. They assembled de novo and compared raw whole genome sequencing data from 23 publicly available genomes of *C. hominis*, *C. parvum*, and *C. meleagridis*. The results of the analysis revealed that most genomes have a size of 9.0 Mb. Only *C. parvum* from humans (UKP14) and two *C. hominis* isolates (SWEH2 and SWEH5) have a genomic size of less than 9.0 Mb. *C. parvum* genome identity within the species, in comparison to the *C. parvum* Iowa type II isolate used as a reference genome, was found to be 99.51–99.93%. The genetic similarities between *C. hominis* genomes were around 96.8%, whereas *C. meleagridis* had a lower global identity of 91.5%. *C. hominis* and *C. meleagridis* had the highest number of single nucleotide variants in coding regions, with more than 150,000 and 400,000, respectively. Single nucleotide variants detection also revealed that *C. hominis* and *C. meleagridis* had the highest number of these, with *C. meleagridis* accounting for 50% of all identified variants. Additionally, the highest number of deletions was found in *C. meleagridis*, followed by *C. hominis*. Although some indel genes have undergone partial annotation, the majority of them still encode uncharacterized proteins, posing a challenge for future research.

The three common intestinal *Cryptosporidium* species in cattle exhibit significant differences in host range, pathogenicity, and public health implications. To gain insight into the genetic determinants of biological differences between *C. bovis* and *C. ryanae*, Xu et al. [9] sequenced their genomes and mapped them to reference *C. parvum* Iowa. Interestingly, the authors proved that *C. bovis* and *C. ryanae* have gene organization and metabolic pathways similar to *C. parvum*. However, their genomes lack invasion-associated mucin glycoproteins, insulinase-like proteases, MEDLE secretory proteins, and other SPDs, which suggests the narrower host range of these two species. Furthermore, the loss of some other SPDs, such as FLGN, SKSR, and NFDQ proteins, might contribute to the reduced pathogenicity of *C. bovis* and *C. ryanae,* which should be further investigated. Comparative genomic analysis of these three intestinal species reveals reductions in secreted pathogenesis determinants in bovine-specific and non-pathogenic *Cryptosporidium* species.

### 2.3. Overcoming Challenges in Isolating Cryptosporidium DNA from Clinical Samples

*Cryptosporidium* is a challenging parasite to study due to the lack of effective in vitro cultivation methods for completing its life cycle. As a result, extracting high-purity DNA from oocysts obtained from fecal specimens remains difficult. In this review, recent advances in methods for isolating and enriching *Cryptosporidium* genomic DNA from clinical specimens for whole-genome sequencing are discussed.

Guo et al. [10] developed a method that combined sucrose and cesium chloride density gradient separation with immunomagnetic separation (IMS) for purifying oocysts from fecal samples of humans and animals with six *Cryptosporidium* species or genotypes. WGS was used to verify the genome coverage and contamination. The results showed that the generated sequence data covered 94.5% to 99.7% of *Cryptosporidium* genomes, with minor contamination from bacterial, fungal, and host DNA. Twenty WGA products with low CT values were submitted to whole-genome sequencing, generating sequence data covering 94.5% to 99.7% of *Cryptosporidium* genomes with mostly minor contamination from bacterial, fungal, and host DNA. These results suggest that this strategy can effectively isolate and enrich *Cryptosporidium* DNA from fecal specimens for whole-genome sequencing.

Anderson et al. [17] performed a simplified version of the above method using IMS combined with PCR to sequence a positive *C. hominis* sample (subtype IbA9G3). The results showed a 91.25% target DNA coverage, with an improvement in parameters compared with the previously published *C. hominis* reference strain TU502.

Similarly, Hadfield et al. [18] developed a preparatory method based on a combination of salt flotation, IMS, and surface sterilization of oocysts prior to DNA isolation directly from human fecal samples. The IMS method was the most efficient and was applied to the sequencing of 17 fecal samples. The results showed eight new whole genome sequences, including two *C. hominis* and six *C. parvum* subtypes. The authors concluded that their method could be useful for clinical sample analysis. To provide a whole perspective of the genomic studies, the following table (Table 2) and figure (Figure 1) summarize the key aspects of the research during the 20 years of *Cryptosporidium* spp. genomics.

## 3. Proteome

The first publications about the *Cryptosporidium* genome presented the possibility of investigating the proteins expressed by this parasite; its proteome can provide complementary information about the biology of this complex organism. However, despite the development of proteomic techniques, one of the significant challenges in studying the proteome of *Cryptosporidium* is the difficulty in cultivating the parasite, resulting in inadequate material for analysis. Accordingly, only a few proteomic analyses have been conducted on *Cryptosporidium* spp., emphasizing the need for more research in this area.

### 3.1. MALDI-MS/MS-Based Proteomics in Cryptosporidium Studies

Mass spectrometry-based proteomics using matrix-assisted laser desorption/ionization tandem mass spectrometry (MALDI-MS/MS) has emerged as a valuable tool in *Cryptosporidium* research. This technique allows for the comprehensive identification and characterization of proteins within the parasite, providing valuable insights into its biology and pathogenicity.

#### 3.1.1. MALDI-MS/MS-Based Proteomics Preceding the Genomic Era of *Cryptosporidium*

Magnuson et al. [24] conducted the first such study before the genomic era of *Cryptosporidium* investigations; in their study, they implemented matrix-assisted laser desorption ionization–time of flight mass spectrometry (MALDI-TOF MS) for proteomic analysis of the parasite. The important effect of these investigations was that they obtained reproducible patterns of spectral markers with increasing sensitivity after lysing the oocysts with a freeze–thaw procedure. However, because of the lack of genomic data, their analysis focused mainly on differentiating between *C. parvum* and *C. muris* based on the spectral peaks patterns specific to the genus. Furthermore, the authors proved that the disinfection of the oocysts resulted in the reduction in and/or elimination of the patterns of spectral markers.

Using the same MALDI-TOF MS approach, Glassmeyer et al. [25] conducted a study seven years later and received spectral peaks specific to the oocyst and as well as sporozoites. In this study, the authors focused mainly on sample preparation, including mixing and standing for 45 min before spotting. As a result, the mass spectra of intact oocysts showed similar peaks to those of sporozoites, indicating that the laser ablates both the oocyst wall and its internal constituents.

#### 3.1.2. Application of MALDI-MS/MS-Based Proteomics for Clinical Samples

Limited data on peptides and proteins from different *Cryptosporidium* isolates, including *C. parvum* and *C. hominis*, is a challenge in the field of *Cryptosporidium* research, mainly because of the absence of readily available animal models. Consequently, proteomic studies have gained significance in analyzing clinical samples. Gathercole et al. [26] developed a refined protocol for purifying oocysts from clinical fecal samples of human patients using salt flotation and potassium bromide density centrifugation. The purified oocysts were then analyzed using matrix-assisted laser desorption/ionization–time-of-flight mass spectrometry (MALDI-TOF MS) to identify specific spectral markers unique to *C. hominis* and *C. parvum*. This study successfully demonstrated the potential utility of the purification method in clinical samples, providing high-quality reference spectra for further peptide and protein discovery and biomarker identification. The results also showed the applicability of MALDI-TOF MS in analyzing *Cryptosporidium* spp. from natural infections, broadening the range of microorganisms that can be studied using mass spectrometry techniques.

### 3.2. LC-MS/MS-Based Proteomics in Cryptosporidium Studies

#### 3.2.1. LC-MS/MS-Based Proteomics: Unveiling Insights into the Biology of *Cryptosporidium*

The availability of the *Cryptosporidium parvum* genome sequence has allowed for large-scale global analyses of its expressed proteome in oocysts and sporozoites. Additionally, the widespread adoption of liquid chromatography coupled with mass spectrometry technology (LC-MS/MS) has facilitated more in-depth protein analysis, opening new avenues for future studies on the biology of *Cryptosporidium* spp. 

In the first such study, Snelling et al. [27] aimed at the known proteome of *C. parvum* in the non-excysted (transmissive) and excysted (infective) forms. The authors used LC-MS/MS with a stable isotope N-terminal labeling strategy to identify and quantify proteins in soluble fractions of both non-excysted and excysted sporozoites. The authors confirmed the expression of many previously hypothesized proteins, as well as the presence of several secreted or surface proteins in sporozoites that are unique to either the Apicomplexa or the *Cryptosporidium* genus. The analysis identified a total of 303 *C. parvum* proteins, of which 26 proteins were found to be significantly upregulated during excystation. These findings suggest that future vaccine development strategies could potentially focus on selecting antigens that block the recognition and attachment of the parasite to host cells, thereby preventing infection. Of particular interest, the analysis revealed the presence of seven proteins that are unique to the genus *Cryptosporidium* and/or the phylum Apicomplexa. Notably, five of these proteins contain a signal peptide at their N-terminus, indicating that they are secreted proteins. These proteins are likely to play a specialized role in the invasion machinery of the parasite, and further investigation into their functions could provide insights into the pathogenicity of the parasite. In addition, the parasite is likely protected against stress and apoptosis by the abundant presence of heat shock proteins (HSP) and ribosomal proteins, which act as protective mechanisms. Three apicomplexan-specific proteins and five *Cryptosporidium*-specific proteins were found to be augmented in excysted invasive sporozoites. The authors also discovered eight promising targets for developing vaccines or chemotherapies that could prevent parasite entry into host cells, e.g., glycosylinositol phospholipids and glycosylphosphatidylinositol protein anchors, which are abundant in the surface membranes and are increasingly recognized as important modulators of the immune function during infection. Moreover, this study provided further evidence of the expression of antigenic proteins, namely Cpa135 (N64) and GP900 (N88). These proteins not only induce the production of specific antibodies but also elicit a cellular Th1 response. Furthermore, the outcome of the studies was the identification of proteins that represent around 8% of the proteome, predicted based on the whole genome sequencing by Andersson et al. [17] and Xu et al. [19].

Sanderson’s paper [28] presents an in-depth analysis of the expressed protein repertoire of *Cryptosporidium parvum*. The authors used three independent proteome platforms (2-DE LCMS/MS, 1-DE LC-MS/MS, and multi-dimensional protein identification technology (MudPIT analysis)) to identify more than 4800 individual proteins representing 1237 nonredundant proteins. The peptide data were mapped to the respective locations on the *C. parvum* genome and a publicly accessible interface was developed for data mining and visualization in CryptoDB. The data provide a valuable resource for improved annotation of the genome, verification of predicted hypothetical proteins, and identification of proteins not predicted by current gene models. The expressed proteome indicates the expression of proteins important for invasion and the intracellular establishment of the parasite, including surface proteins, constituents of the remnant mitochondrion, and apical organelles. Proteomic analysis of the *C. parvum* genome revealed that between 25% and 40% of the potential gene products were classified as “hypothetical proteins” due to a lack of sufficient similarity to known proteins. However, the study provides proteomic evidence for 482 of these hypothetical proteins in *C. parvum*, which represents 39% of the total proteome detected. It is estimated that 69%-74% of the coding region of the *C. parvum* genome contains genes based on imperfect algorithmic prediction programs. Furthermore, the study suggests that an additional 72 regions of the genome scaffold may contain coding exons that were not previously recognized by current gene prediction models. In this study, the dataset of transporter-like proteins associated with the oocyst/sporozoite life- cycle stages have been expanded to include 64 proteins. The proportion of the total proteome that these proteins represent is estimated to be 9%, which is consistent with the prediction based on genome analysis. It is worth noting that, despite using such advanced techniques, identified proteins corresponded to 30% of the predicted proteome, which highlights the current limitations in understanding *Cryptosporidium* biology.

Another example of a proteomic study aimed at exploring the stage-specific proteome of *Cryptosporidium parvum* is the sporozoite proteome analysis of *C. parvum* [29]. The authors paid particular attention to the presence of *C. parvum* hydrophobic and membrane proteins. Therefore, the study aimed to reduce sample complexity using protein fractionation, enabling the detection of proteins present in lower abundance in a complex protein mixture. In this study, therefore, they used SDS–PAGE to fractionate the sporozoite protein of *C. parvum*, followed by an LC-MS/MS approach. A total of 135 protein hits were recorded from 20 gel slices, with many hits occurring in more than one band. Excluding non-*Cryptosporidium* entries and proteins with multiple hits, 33 separate *C. parvum* entries were identified. These included structural, metabolic, and hypothetical proteins covering a wide range of pHs and molecular weights. Several metabolic enzymes, including the glycolytic enzyme glyceraldehyde-3-phosphate dehydrogenase and protein disulfide isomerase, were identified in this study. These findings support the hypothesis that glycolysis is the primary energy source in *C. parvum* and that the parasite relies mainly on the anaerobic oxidation of glucose for energy production. The presence of these enzymes in the proteome is notable and provides further evidence of the reliance of the parasite on glycolysis.

#### 3.2.2. LC-MS/MS-Based Proteomics: Characterizing Specific *Cryptosporidium* Fractions

With the development of mass spectrometry as a protein discovery tool, studies on the *Cryptosporidium* proteome have become more specific and focused on characterizing specific fractions of the parasite. The first study using mass spectrometry aimed at identifying the components of the oocyst wall and affinity purification to isolate glycoproteins from the sporozoites in order to investigate how sporozoites are tethered to the oocyst wall [30]. The investigation revealed that COWPs comprise 75% of oocyst wall proteins, with COWP1 being the most abundant. COWP8 and COWP6 were also present. Affinity-purified glycoproteins included known mucin-like glycoproteins, a GalNAc-binding lectin, and novel glycoproteins, suggesting their involvement in tethering sporozoites to the oocyst wall. These results suggest that mucin-like glycoproteins may contribute to the fibrils and/or globules that tether sporozoites to the inner surface of oocyst walls.

Another examination focused on organelles called rhoptries and their contents, which are crucial in establishing productive infection during invasion [31]. In order to identify rhoptry proteins and explore the parasite’s invasion pathway and pathogenic mechanisms, subcellular fractionation and mass spectrometry analysis were used. The authors detected 22 potential novel rhoptry proteins by fraction analysis using LC-MS/MS and online softwares: MASCOT in the NCBI, CryptoDB v5.0, and EupathDB v2.1. It is worth noting that these novel candidates could serve as targets for future research on the *C. parvum* invasion pathway and the function of rhoptry proteins. 

#### 3.2.3. LC-MS/MS-Based Proteomics: Host–Parasite Interactions

In recent years, studies have mainly focused on identifying and characterizing proteins involved in the host–parasite interaction, elucidating the parasite’s virulence mechanisms, and discovering potential drug targets for the development of effective therapeutic agents. Li et al. [32] investigated the excystation process in *Cryptosporidium andersoni* by collecting and purifying oocysts from naturally infected adult cows. Using both immunological and molecular methods, this study identified several proteins as putative virulence factors. Their study yielded interesting data; they identified 1586 proteins in *C. andersoni* and found 17 differentially expressed proteins (DEPs) after excystation, including 10 upregulated and 7 downregulated proteins. This is a larger number of proteins than previously reported for *C. parvum*, suggesting that *C. andersoni* may have a more complex proteome or different proteins involved in excystation. This study identified several proteins as putative virulence factors using both immunological and molecular methods. For instance, serine protease and aminopeptidase are associated with excystation, while P23 and P30 are involved in adhesion. TRAP and thrombospondin-related anonymous proteins are implicated in parasite gliding and cell penetration. Cp2, Cap135, and secreted phospholipase are associated with invasion, while HSP70 and HSP90 are involved in stress protection. These proteins play key roles in various stages of the *C. parvum* life cycle and are potential targets for future drug and vaccine development. These findings provide new insights into the protein composition of *C. andersoni* oocysts and potential excystation factors, which could be useful in identifying genes for diagnosis, vaccine development, and immunotherapy for *Cryptosporidium* spp.

In 2021, Kacar et al. [33] conducted a study to investigate the effectiveness of proteome method analysis on the sera of Holstein calves naturally infected with *Cryptosporidium* spp. after being treated with high-quality colostrum and paromomycin. The study analyzed sera samples collected at the 0th and 3rd days post-treatment across three different groups of animals: the first group received only paromomycin (PC); the second group received paromomycin and colostrum (PCOL); the third group received paromomycin, colostrum, and sodium bicarbonate (PBCOL). The study’s results indicated that colostrum treatment had the most positive effect, as significant changes in several proteins were observed. These proteins played important roles in binding/transporter, catalytic activity, regulation of molecular functions, and regulation of structural–molecular activity. Additionally, the study revealed the down-regulation of many inflammatory mechanisms and processes in the colostrum-treated groups, which provides insights into the mechanism of action of colostrum in treating the infection.

#### 3.2.4. LC-MS/MS-Based Proteomics: Drug Targets and Diagnosis

Unfortunately, there is currently no fully effective drug or vaccine available to treat cryptosporidiosis, despite the reemergence of *Cryptosporidium* infection. A large-scale label-free proteomics approach was employed to understand the detailed interaction between the host and *Cryptosporidium parvum* [34]. Among the 4406 proteins identified, 121 proteins were differentially abundant (>1.5-fold cutoff, *p* < 0.05) in *C. parvum*-infected HCT-8 cells compared with uninfected cells. Specifically, 67 proteins were upregulated, and 54 proteins were downregulated 36 h post-infection. Analysis of the differentially abundant proteins revealed that the host cells mounted an interferon-centered immune response against *C. parvum* infection and that the parasite extensively inhibited metabolism-related enzymes in the host cells. Several proteins were further verified using quantitative real-time reverse transcription polymerase chain reaction and Western blotting. This systematic analysis of the proteomics of *C. parvum*-infected HCT-8 cells identified a range of functional proteins that participate in host anti-parasite immunity or act as potential targets during infection, providing new insights into the molecular mechanism of *C. parvum* infection.

Karpe et al. [26] investigated the biochemical interactions underlying *Cryptosporidium parvum* infection in C57BL/6J mice. To do so, they collected various specimens from the mice 10 days post-infection, including fecal samples, blood, liver tissues, and luminal contents. They then analyzed the proteomes and metabolomes of these specimens using high-resolution liquid chromatography and low-resolution gas chromatography coupled with mass spectrometry. Their aim was to shed light on the biochemical mechanisms of cryptosporidiosis, which could have implications for clinical diagnosis. Univariate and multivariate statistical analysis revealed altered host and microbial energy pathways during infection, including depleted glycolysis/citrate cycle metabolites, increased short-chain fatty acids and D-amino acids, an abundance of bacteria associated with a stressed gut environment, and upregulated host proteins involved in energy pathways and *Lactobacillus* glyceraldehyde-3-phosphate dehydrogenase. Liver oxalate also increased during infection. They observed that microbiome–parasite relationships were more influential than the host–parasite association in mediating major biochemical changes in the mouse gut during cryptosporidiosis. Defining this parasite–microbiome interaction is the first step toward building a comprehensive cryptosporidiosis model to discover biomarkers and develop rapid and accurate diagnostics. Table 3 and Table 4 offer a detailed overview of *Cryptosporidium* proteomics studies, encapsulating critical information about the specimens, applied techniques, and search engines utilized, providing a comprehensive context to the review.

## 4. Transcriptome

Studying the transcriptome of *Cryptosporidium* has been challenging due to the inability to culture the parasite through its complete life cycle and the lack of suitable host cells. Additionally, interpreting transcriptomes is complicated by the asynchronous life cycle and limited genome annotation. Recent progress in the survey of *Cryptosporidium* spp. has improved the tools, including characterizing relevant life cycle stages and host cells, offering the potential for enhanced understanding and therapeutic advancements.

### 4.1. Hybridization-Based Characterization of the Cryptosporidium Transcriptome

Microarrays are widely used for gene expression analysis, SNP identification, and genome comparisons. They contain thousands of DNA probes attached to slides, representing mRNA sequences from the target species. The probes can be oligonucleotides, cDNA, or PCR products. Hybridization to cDNA arrays is more specific but costlier than oligonucleotide arrays [36].

One of the most important hybridization-based studies was the development of the first Agilent microarray for *C. parvum* (CpArray15K) in combination with real-time PCR, which covered all predicted ORFs in the parasite genome founding 2000 upregulated genes, of which more than 50% of the genes were hypothetical. This method allows for the comparison of gene expression profiles between untreated and UV-irradiated oocysts to identify genes involved in the response to environmental stress. During the oocyst stage, parasites show high protein synthesis activity, indicated by increased transcript levels of genes involved in ribosome biogenesis, transcription, and translation. Furthermore, the authors observed a significant upregulation of genes related to ubiquitin/proteasome-mediated protein degradation and posttranslational modification, suggesting the recycling of proteins as a strategy to compensate for the parasite’s inability to synthesize amino acids endogenously. A significant finding of these studies was the discovery that energy metabolism in oocysts is characterized by the notable expression of the lactate dehydrogenase (LDH) gene. Given the association of LDH with the parasitophorous vacuole, targeting the enzyme could offer therapeutic advantages in drug development. Furthermore, the study showed how stress-related genes, such as those in the TCP-1 family and certain thioredoxin-associated genes, play significant roles in oocysts’ recovery from UV-induced damage, which provides valuable insights into the behavior and response mechanisms of parasites [37].

In an in vitro model of human intestinal cryptosporidiosis, the authors made a notable observation regarding certain *C. parvum* RNA transcripts. Specifically, these transcripts were found to be transported into the nuclei of host epithelial cells during infection. One of the identified transcripts, Cdg7_FLc_0990, showed a unique nuclear delivery facilitated by the heat-shock protein 70-mediated nuclear import mechanism. Moreover, when Cdg7_FLc_0990 was overexpressed in intestinal epithelial cells, it led to significant changes in the expression levels of specific genes. These changes were similar to the alterations observed in host cells after *C. parvum* infection. These intriguing findings suggest that *C. parvum* transcripts with low protein-coding potential are selectively delivered to infected host cells, and they may have a role in influencing gene transcription within these cells [38].

Sawant et al. [39] analyzed the transcriptomes of *C. parvum*-infected ileocecal regions of mice developing tumors to explore the signaling pathways potentially involved in the development of neoplasia induced by parasites. The authors observed, for the first time, downregulation of the expression of β-defensin, an anti-microbial target of the parasite, in response to *C. parvum* infection in the transformed tissues. This phenomenon has been speculated to result from *C. parvum*’s resistance to host defense, possibly due to the upregulated expression of interferon-stimulated genes. Guanylate binding proteins (GBP2, GBP4, GBP6, GBP8, and GBP11), a superfamily of large GTPases, were observed to be upregulated. Their upregulation is known to be induced by IFN-γ as a host response to external pathogens. Furthermore, the identification of immune suppressor cells and accumulation of pro-inflammatory mediators suggests that chronic inflammation induced by persistent *C. parvum* infection assists in the development of an immunosuppressive tumor microenvironment.

### 4.2. Reverse-Transcription PCR (RT-PCR)-Based Transcriptomics

One of the most significant and time-consuming studies in *Cryptosporidium* biology was the analysis of transcriptomes over a 72-h in vitro infection of human epithelial cells with *C. parvum*. Transcript abundances of 3302 genes (representing 87% of the genome) were determined using RT-PCR in a temporal study. Gene expression patterns, normalized to 18S rRNA levels, revealed distinct profiles associated with different developmental stages. Notably, metabolic, ribosomal, and proteasome protein expression did not show congruence with 18S rRNA levels. Distinct transcripts were increasingly detected from 6 h post-infection, with a significant portion of genes showing the highest expression at either 48 or 72 h. Cluster analysis of the complete dataset revealed nine distinct clusters, indicating that gene expression is independent of chromosomal location. Transcription-related genes peaked at 2 h, translation-related proteins dominated at 6 h, and structural genes like myosin and tubulin were most prevalent at 12 h post-infection. Transcript abundance increased during the parasites’ sexual reproduction from 36 to 72 h. These findings suggest a more intricate system of transcription regulation in *C. parvum* than previously believed [40].

### 4.3. Sequenced-Based Characterization Cryptosporidium spp.

#### 4.3.1. Host–Parasite Interaction

To uncover critical parasite pathways involved in pathogenesis, it is crucial to investigate the gene expression profile of *C. parvum* during infection. Mirhashemi et al. [41] compared gene expression in infected and uninfected monolayers of cultured cells and identified genes involved in apoptosis, cell growth, and cell division. They demonstrated that these genes exhibited differential expression in infected cells. The study revealed a total of 696 genes with altered expression. The results of the current study indicated upregulation of host heat shock genes and genes encoding Cysteine-X-Cysteine (CXC) chemokines. Host genes involved in cell proliferation and apoptosis showed differential expression. However, RNA-seq data of infected pig intestinal monolayer cells suggested no changes to stress or apoptosis-associated host genes, indicating the host was not much affected by the parasite infection.

Matos et al. [42] focused on analyzing the transcriptome of *C. parvum* in oocysts, sporozoites, and infected cell monolayers at different time points ranging from 2 to 48 h post-infection for a better understanding of gene regulation in the parasite life cycle. This comparative study revealed significant differences between the transcriptomes expressed outside and inside the host. Extracellular stages (oocysts and sporozoites) express similar transcriptomes both in terms of diversity and function, while the intracellular transcriptome shows a significant enrichment of functions related to ribosomes and protein synthesis. Furthermore, it is worth noting that the study confirmed high LDH mRNA abundance in oocysts, as reported previously [43]. Interestingly, among 21 genes encoding enzymes in the glycolytic pathway, only LDH mRNA is expressed at such a high level. However, the functional significance of this observation requires further investigation. One general limitation of the study is related to the existing methods to culture *Cryptosporidium* parasites, which result in a patchy infection with many host cells remaining uninfected.

One of the mysteries related to *Cryptosporidium* biology is the way this parasite regulates its genes during developmental transitions. For this purpose, Li et al. [44] provide an initial characterization of the *C. parvum* non-coding transcriptome to facilitate further investigations into the role of putative long non-coding RNAs (lncRNAs), which play a key role in the pathogenesis of many diseases. The authors utilized a comprehensive set of stranded RNA-seq data from 151 diverse lifecycle stages, covering both asexual and sexual stages. They systematically analyzed long non-coding RNA (lncRNA) characteristics, conservation, expression profiles, and their relationship with neighboring genes. The results of the study revealed that for the presence of 396 high-confidence lncRNAs, 363 occur as antisense transcripts to mRNAs, and 33 are 455 encoded in intergenic locations. The authors propose that lncRNAs play a role as a mechanism of parasite-induced host transcriptome regulation.

To understand the molecular interactions between human ileocecal adenocarcinoma (HCT-8) cells and *Cryptosporidium* species, the researchers conducted a comprehensive transcriptomic analysis. They explored the expression profiles of messenger RNAs (mRNAs), long non-coding RNAs (lncRNAs), microRNAs (miRNAs), and circular RNAs (circRNAs) at 3 and 12 h post-infection. The functional predictions using GO and KEGG pathway analyses revealed significant roles for the differentially expressed RNAs during *C. parvum* infection. Moreover, the ceRNA regulatory network demonstrated how a single lncRNA or circRNA can influence multiple miRNAs, which, in turn, co-regulate additional mRNAs. Notably, the downregulation of hsa-miR-324-3p and hsa-miR-3127-5p seems to play a key role in regulating circRNAs, lncRNAs, and mRNAs. These findings provide valuable insights into the response of human intestinal epithelial cells to *C. parvum* infection and provide a basis for further research [45].

The latest transcriptomic analysis focused on understanding the expression patterns of 144 human and *Cryptosporidium* genes at various stages of the pathogen’s life cycle. To achieve this, the authors employed COLO-680N cells and utilized flow cytometry and microscopy along with the *C. parvum*-specific antibody Sporo-Glo™ to characterize infected cells 48 h after infection with either *C. parvum* or *C. hominis*. Interestingly, *C. parvum*-infected cells showed higher signal levels using Sporo-Glo™ compared to *C. hominis*-infected cells, likely due to the specificity of Sporo-Glo™ towards *C. parvum*. This enabled the detection of infected host cells without the need for fluorescent labeling strategies, significantly enhancing the study’s accuracy. The authors then employed NanoString transcriptomic analysis to assess the expression of genes in the COLO-680N cell line in comparison with sporozoites of both *C. parvum* and *C. hominis*. The raw gene expression counts in all infected cell cultures revealed significantly low levels of *Cryptosporidium*-specific genes, almost reaching the detection limit of the nCounter^®^. In contrast, all parasite genes were expressed at higher levels in sporozoites when compared to the infected cells [46].

#### 4.3.2. Drug and Therapeutic Targets

One of the main applications of transcriptomic research is the identification of potential drug targets. Currently, there are no definitive treatments (vaccines or chemoprophylaxis) or control measure options available to fully treat cryptosporidiosis or prevent infection in both humans and animals. Therefore, Lippuner et al. [43] employed RNA-Seq to analyze and compare purified *C. parvum* sporozoites from oocysts, the intestine of infected calves, or infected HCT-8 cells. Their aim was to identify key genes for the parasite’s life cycle and potential drug/vaccine targets. After in vitro parasite infection of the human ileocecal adenocarcinoma (HCT-8) cells, RT-PCR was used to assess the temporal gene expression patterns. Two hours post-infection at the early trophozoite stage, sugar-nucleotide transporter genes, transcription, and DNA-associated genes were upregulated, compared to sporozoites alone which indicated that the parasite obtains nutrition from the host. Mucins, known to be involved in host cell attachment, were also upregulated in vivo. Genes encoding oocyst wall proteins were upregulated in both in vitro and in vivo infection compared to sporozoites. In this study, 3774 protein-encoding transcripts were identified in *C. parvum*. Of these, 173 genes were upregulated in sporozoites, including 26 coding for predicted secreted proteins. Conversely, 1259 genes were upregulated in intestinal stages (merozoites/gamonts), with enrichment in 63 biological processes and upregulation in 117 genes across 23 metabolic pathways. Transcriptomic analysis proved that genes associated with biosynthetic processes, such as mucin (gp40/15 and gp900), were upregulated in intracellular stages. Additionally, specific genes, like calcium-dependent protein kinases (CDPKs), were found to be expressed across all stages of the parasite, making them potential drug targets against the parasite. However, a minor limitation of this study is that it did not explore transcriptomic changes in host cells.

Nitazoxanide exhibits antiparasitic efficacy in healthy individuals; however, it is often ineffective in individuals with compromised immune systems. In light of this, Gallego–Lopez conducted a study on soluble Toxoplasma gondii antigens (STAg) as a potential therapeutic target against *C. parvum* infection. The inspiration for this study was the fact that previous studies have shown that animals with chronic diseases are protected against lethal secondary infections from other parasites. The authors demonstrated that STAg treatment effectively decreased *C. parvum* infection in IFN-g-KO mice and conducted a transcriptomic analysis to understand this novel IFN-g-independent mechanism. When comparing infections in tissues treated with PBS and STAg, IFN-g-KO mice showed a higher number of significant DEGs, indicating that the treatment alone may not be enough to induce host gene expression changes. It appears that *C. parvum* infection collaborates with STAg treatment to trigger the type I interferon immune response. However, the authors did not observe significant differences in transcript abundances when comparing STAg and PBS treatments. One possible explanation is that STAg treatment may specifically affect the host transcriptome while leaving the parasite transcriptome unaffected. The lack of differential expression in *C. parvum* could also be attributed to the limitations in coverage and sequence depth in their study [47]. 

Table 5 offers a detailed overview of *Cryptosporidium* transcriptomics studies.

## 5. Discussion

Here, we have reviewed promising advancements in genomic, proteomic, and transcriptomic studies in the field of *Cryptosporidium* spp. The current availability of reference genomes for *Cryptosporidium* spp. has not only impacted our understanding of the basic biology of the parasite but also caused delays in diagnostics and therapeutics. For instance, the first published data of sequencing *C. parvum* and *C. hominis* served as an excellent basis for further analyses. Abrahamsen et al. [11] and Xu et al. [19] highlighted that the metabolic/biosynthetic pathways of this microorganism present numerous novel avenues for drug targets and drug discovery. The continuation of this discovery was carried out in studies conducted by Umejiego et al. [48], using in silico docking experiments and high-throughput screening. They identified several compounds (Compounds “A”, “F”, “G”, and “H”) with specific efficacy against *Cryptosporidium* spp. in cell culture, surpassing the performance of paramomycin. These findings offer promising leads for potential drug development against *Cryptosporidium* spp., addressing the challenge of limited biosynthetic capabilities in these parasites and their resistance to traditional treatments. Furthermore, transporter proteins and a mitochondrial-associated alternative oxidase have been identified as potential drug targets in *Cryptosporidium* spp. These proteins play crucial roles in salvaging nucleotides and essential amino acids from the host, and the mitochondrial-associated alternative oxidase is responsible for catalyzing the reduction of oxygen to water without generating harmful free radicals or oxidizing agents (such as superoxide or H_2_O_2_) [49]. Targeting these proteins could offer promising avenues for drug development against *Cryptosporidium* spp., addressing its unique metabolic features and reducing the reliance on host resources for survival. For many *Cryptosporidium* genomes, significant time has passed since their initial sequencing and assembly. These references may require updated sequencing using next-generation technologies and genome re-annotation and polishing to ensure they do not lead to misleading interpretations in genetic and molecular studies. The latest sequencing methods hold great promise for such studies. Single-cell sequencing provides hope for a deeper comprehension of the parasite genome, particularly in cases of mixed infections that are common in humans from endemic regions, where multiple parasite species coexist in a single host. However, while this method helps address challenges related to mixed populations, it also enables the detection of low-frequency variations [21]. Genetic recombination during sexual replication generates diversity that affects virulence and transmissibility. The establishment of clonal cultures of isolates would help overcome some of the challenges associated with analyzing this diversity. In the case of *Cryptosporidium* spp., long-read sequencing offers a number of advantages over short-read sequencing. While short reads can capture the majority of genetic variation, long-read sequencing allows the detection of complex structural variants that may be difficult to detect with short reads. In addition, short reads do not provide information about concurrent alternative or rare splicing isoforms [50].

Long-read technologies such as PacBio IsoSeq and ONT direct RNAseq can be used to study alternative splicing, facilitating gene verification and annotation, as well as identifying novel splicing forms. Furthermore, there are currently no models for nucleotide modification in *Cryptosporidium* species, and reports on methylation sites in the genome are mixed [51,52], which constitutes one of the challenges for the future of *Cryptosporidium* spp. genomic research. Additionally, it is crucial to widely implement the integration of both short-read and long-read sequencing approaches in *Cryptosporidium* spp. genomic research, as this is particularly helpful in the process of genome annotation. Re-annotations are particularly important for *C. hominis* as another significant causative agent of human disease and for *C. tyzzeri* as a murine model of human infection, both of which require urgent re-annotation.

The use of proteomics techniques was significant in the studies of *Cryptosporidium* spp. to investigate its most intriguing biological questions, such as the control of *Cryptosporidium* spp. development through its life cycle and its interactions with the host environment. However, unlike other apicomplexan parasites, relatively few proteomic analyses have been conducted for *Cryptosporidium* spp. so far. Due to the challenges in cultivating these parasites, most proteomic studies have been limited to the stages of oocyst and sporozoite of *C. parvum*. As a result, there are many aspects that need clarification in *Cryptosporidium* proteomics, including potential drug targets and the parasite–host relationship. Snelling et al. [27] conducted a successful quantitative analysis of proteins identified in both parasites released from oocysts and those that did not undergo excystation. They identified a total of 26 proteins in the soluble fractions that showed significantly higher expression levels after excystation compared to unexcysted oocysts. Furthermore, the authors demonstrated that 3 out of the 26 proteins were specific to Apicomplexa, and 5 were specific to *Cryptosporidium* spp. According to the authors, all these proteins are involved in the pathogenesis. Nevertheless, it remains to be determined whether *Cryptosporidium* spp. elicits the same response when it comes into contact with the host or when it is internalized. On the other hand, analyses conducted by Li et al. [34] revealed that inhibiting the expression of SHMT2 by *C. parvum* not only resulted in the blockage of the host’s metabolism but also indicated a potential strategy employed by the parasite to secure the supply of essential nutrients. Previous studies have shown the promotion of exogenous purine nucleosides during *C. parvum* infection [53], while genomic analysis indicated the loss of de novo pyrimidine synthesis in *C. parvum*. This makes the parasite almost entirely dependent on importing these compounds from the host to meet its purine and pyrimidine requirements. Based on these processes, *C. parvum* would acquire an ample supply of purines and pyrimidines, ensuring successful parasitism and replication. This might be achieved by potentially interfering with the host’s purine and pyrimidine metabolism. These studies provided new knowledge, but for a better understanding of these mechanisms, deeper investigations are necessary.

Analysis of the mRNA transcriptome and protein expression have already brought and will continue to provide important insights into the transcriptional landscape of the *Cryptosporidium* spp. genome. Studying full transcripts can help enhance our understanding of the content of UTRs; for example, it helps us identify promoter sequences that can be used for genetic studies and may serve as potential therapeutic targets. In recent years, significant technological advances have led to the availability of RNA-Seq, which is highly sensitive, with low background noise and a broad dynamic range, enabling quantitative analysis of transcripts present at both low and high levels. In the case of *Cryptosporidium* spp., transcriptomic studies have greatly contributed to understanding the host–parasite interactions. RNA-seq data from infected pig intestinal monolayer cells suggested no changes in host stress or apoptosis-related genes, indicating that the host was not significantly affected by the parasite infection [41]. Furthermore, microarray studies on long non-coding RNAs (lncRNAs) have shown an upregulation of inflammatory factors (TNF and interleukins), cell proliferation factors, the Wnt signaling pathway, the hedgehog pathway, and tight junction-associated proteins [54]. Recent studies have demonstrated the production and transport of lncRNAs into the host cell nucleus during infection. One such transcript of *C. parvum*, Cdg7_FLc_0990, interacts with the host cell chaperone heat shock protein 70 (hsp70) to enter the host nucleus. These transcripts affect the host cell transcriptome by upregulating pro-inflammatory genes and hijacking the host histone regulation system [38]. Other putative lncRNAs have also been reported, indicating that lncRNAs are a mechanism of parasite-induced host transcriptome regulation [44]. For RNA-Seq analysis to yield optimal value, a sufficiently high proportion of host cells must be infected to detect transcriptomic changes in the host. Similarly, to accurately quantify changes in the parasite’s transcriptome, the proportion of parasite transcripts to host transcripts must reach a threshold in the total RNA pool being analyzed. These challenges can be addressed by implementing single-cell RNA-Seq and increasing sequencing depth or pre-sequencing enrichment of parasite transcripts, or a combination of both approaches. By employing such methods, RNA-Seq-based investigations offer a unique opportunity for scientists to explore unknown genes or pathways that are crucial for the parasite’s survival, thus presenting potential drug targets.

## 6. Future Perspectives

Despite significant progress in understanding *Cryptosporidium* spp. and its biology using omics techniques, numerous areas remain unexplored. The cornerstone for advancing omics research on *Cryptosporidium* spp. lies in developing an effective cultivation method to obtain a sufficient amount of material for comprehensive analyses. Additionally, it is of utmost importance to refine routine transfection of *Cryptosporidium* spp. and establish a long-term cultivation platform. These objectives are critical to achieving the primary research goals, which are to identify crucial functional genes, understand their roles, and uncover potential therapeutic and vaccine targets. Furthermore, genomics, proteomics, and transcriptomics studies face a significant challenge in uncovering the molecular basis of life cycle changes in *Cryptosporidium* spp.

Addressing these crucial knowledge gaps in *Cryptosporidium* spp. research necessitates the integration of various omics approaches, as well as computational and statistical methods. This strategic integration will pave the way for advanced studies in biology and host–parasite interactions, facilitating a comprehensive analysis of the relationships between hosts and parasites. Consequently, this comprehensive approach will provide a deeper understanding of the mechanisms used by parasites to evade host immune defenses and the host’s balancing mechanisms against parasites’ actions. Departing from a reductionist approach, which analyzes individual components to infer the behavior of complex systems, is pivotal in this endeavor.

## Figures and Tables

**Figure 1 ijms-24-12867-f001:**
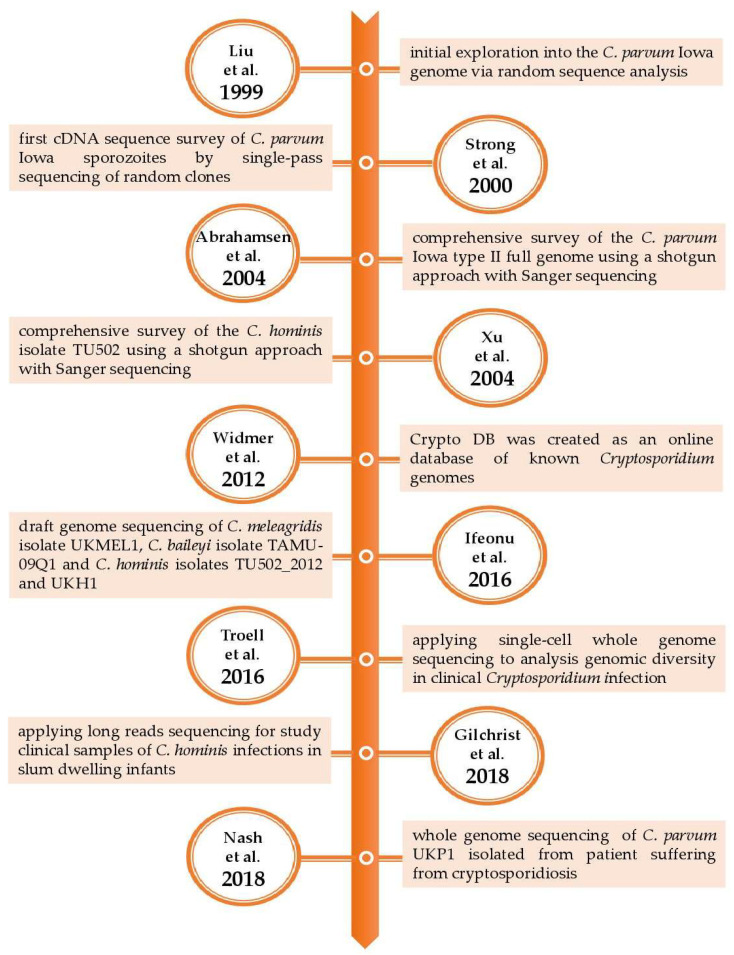
Charting the course of *Cryptosporidium*: a timeline of whole genome sequencing studies [6,7,8,11,13,19,20,21,22].

**Table 1 ijms-24-12867-t001:** Genomic sequencing data across multiple platforms for various *Cryptosporidium* species.

Species	Genomes ID	Platform	Genome Size (Mbp)	# of Contigs	# of Reads	Contig N50 (bp)	Average Coverage
** *C. hominis* **	TU502 237895	Sanger Dideoxy Sequencing	8.70	1422	-	48,000	12
	TU502 2012	Illumina MiSeq	9.10	119	1,810,060	238,509	96
	30976	Illumina Genome Analyzer IIx 100 bp paired-end	9.05	53	35,360,353	470,636	511
	37999	Illumina Genome Analyzer IIx 100 bp paired-end	9.05	78	16,569,87	406,678	367.4
	33537	454 GS-FLX Titanium	9.60	1464	1,157,140	27,749	31
	30974	454 GS-FLX Titanium	8.84	443	1,048,412	78,110	43
	SWEH2	Ion Torrent	8.81	1629	1,791,829	9465	35.2
	SWEH5	Ion Torrent	8.82	1342	2,058,197	14,514	42.4
	UdeA01	Illumina MiSeq	9.04	8	1,080,44	1,103,974	53.4
	UKH1	Illumina MiSeq	9.14	156	3,798,205	179,408	197.3
	UKH3	Illumina MiSeq	9.07	179	1,238,762	167,737	35.8
	UKH4	Illumina HiSeq	9.39	2164	11,895,367	48,766	321.5
	UKH5	Illumina HiSeq	9.06	526	12,649,912	81,885	362.2
** *C. parvum* **	Iowa II 5807	Sanger Dideoxy Sequencing	9.10	18	-	1,014,526	13
	UKP1	Illumina HiSeq	8.88	14	26,000,000	1,092,230	600
	31727	Illumina Genome Analyzer IIx 100 bp paired-end	9.08	337	13,074,496	76,396	116.0
	34902	Illumina Genome Analyzer IIx 100 bp paired-end	9.11	1076	18,907,631	21,594	168.7
	35090	Illumina Genome Analyzer IIx 100 bp paired-end	9.04	3256	14,188,762	4248	3.256
** *C. baileyi* **	TAMU-09Q1	gDNA Illumina library fragment size (bp) 654	8.43	145	6,240,960	203,018	70.06
** *C. muris* **	5808	4.5× Sanger and 10× 454	9.25	97		520,347	10
** *C. chipmunk* ** **genotype I**	1280935	Illumina Genome Analyzer IIx 100 bp paired-end	9.05	50	9,509,783	117,886	200
** *C. bovis* **	310047	Illumina HiSeq 250 bp paired-end	9.11	59	7,080,000	444,382	196
** *C. ryanae* **	515981	Illumina HiSeq 250 bp paired-end	9.06	100	5,130,000	231,122	142.5
** *C. meleagridis* **	UKMEL1	Illumina MiSeq	8.9	57	11,431,022	322,908	110.4

**Table 2 ijms-24-12867-t002:** Timeline and progression of genomic research in *Cryptosporidium* species.

Year	The Greatest Milestone	Genomic Approach	Outcome of Study	Reference
1999	Initial genomic exploration into *C. parvum* Iowa strain	Random sequence analysis	✓ Sequence survey comprising only 2.5% of the *C. parvum* genome ✓ The disadvantage of the method is the limited availability of gene sequences in the public databases at the time	[13]
2000	First cDNA sequence survey of *C. parvum* Iowa oocysts/sporozoites	Random sequence analysis with GSS approach to gene discovery	✓ First broad-based molecular views into the basic biology and cellular metabolism of this experimentally intractable apicomplexan parasite	[6]
2004	Complete genome sequencing of *C. parvum* Iowa type II strain	Whole-genome with shotgun Sanger sequencing	✓ Whole genome was sequenced from a plasmid insert library ✓ “HAPPY” map was used to create a scaffold and order the sequence contigs	[11]
2004	Complete genome sequencing *C. hominis* TU502	Whole-genome with shotgun Sanger sequencing	✓ Whole genome was sequenced from a plasmid insert library ✓ “HAPPY” map was not available; the authors constructed large (~7–8-fold coverage) bacterial artificial chromosome (BAC) libraries for this species and used the *C. parvum* “HAPPY” map to guide assembly	[19]
2012	Comparative genome analysis of two *C. parvum* isolates (TU114 and *C. parvum* Iowa)	Whole-genome sequencing	✓ Small number of highly diverged genes ✓ Transporter genes’ over-representation indicates species-specific infection ability	[12]
2015	Sequencing of genomes *C. chipmunk* genotype I	Whole genome sequencing	✓ Subtyping tool based on two markers for genetic characterization of *C. chipmunk* genotype I was developed ✓ *C. chipmunk* genotype I isolates from humans and wildlife are genetically similar	[23]
2015	Comparative genome analysis of *C. hominis* and *C. parvum*	Whole genome sequencing	✓ Occurrence of genetic recombination in virulent ✓ *C. hominis* subtypes and telomeric gene duplications in *C. parvum* ✓ Sequence similarity and recombination in the gp60 region indicate a potential role in the emergence of highly transmissible *C. hominis* subtypes	[14]
2016	Sequencing of genomes: *C. meleagridis* UKMEL1, *C. baileyi* TAMU-09Q1 and *C. hominis* TU502_2012 and UKH1	Draft genome sequencing	✓ The genome assembly of *C. hominis* is significantly more complete and less fragmented than the previous version ✓ The first versions of genome sequence assemblies and annotations for each isolate	[7]
2016	Genome sequencing of *Cryptosporidium* spp. in clinical samples	Single-cell sequencing	✓ Workflow for whole genome sequencing of single cells of the parasite ✓ Combining sequence data from all single-cell genomes, almost the entire reference genome (99.7%) was accounted for, and most of this (98.8%) was described at > 20× coverage	[21]
2017	Sequencing of the genomes of two specimens of *C. parvum* form China and Egypt	Whole genome sequencing	✓ Differences in subtelomeric gene families, such as SKSR, MEDLE proteins, and insulinase-like proteases, found between sequenced and reference genomes ✓ Most polymorphic genes between genomes mainly encode invasion-related mucin proteins and other secretory protein families	[15]
2018	Analysis of genetic diversity of *C. hominis* infections in slum-dwelling infants in Bangladesh	Long-read resequencing	✓ High rates of sexual recombination and regions of the genome that were highly polymorphic, suggesting areas under selection	[22]
2018	Analysis of a zoonotic isolate of *C. parvum* UKP1 isolated from a person with cryptosporidiosis	Draft genome sequencing	✓ Sequences needed to identify markers important in distinguishing routes of transmission and potential virulence traits for better epidemiological analysis and risk assessment	[8]
2020	Comparative analysis of *Cryptosporidium* species that infect humans	Whole-genome sequencing	✓ Synonymous single nucleotide variants were the most common in *C. hominis* and *C. meleagridis*, while in *C. parvum*, they accounted for around 50% of the SNV observed	[16]
2020	Sequencing of the genomes of *C. bovis* and *C. ryanae*	Whole-genome sequencing	✓ The genome of *C. bovis* has a gene content and organization more similar to *C. ryanae* than to other *Cryptosporidium* species sequenced to date	[9]

**Table 3 ijms-24-12867-t003:** Chronological Advancements in Proteomic Analysis of *Cryptosporidium* Species.

Year	The Greatest Milestone	Outcome of Study	Reference
2000	Initial proteomic study of whole and freeze–thawed *C. parvum* oocysts and freeze–thawed *C. muris*	✓ Identify spectral peaks that can distinguish *Cryptosporidium* at the genus level, as well as specific peaks that enable differentiation between *C. parvum* and *C. muris* oocysts ✓ The pioneering utilization of MALDI-TOF peptide mass fingerprinting (PMF) for *Cryptosporidium* oocysts analysis	[24]
2007	Proteomic analysis of *C. parvum* oocysts	✓ Improvements in the sample preparation before analysis using MALDI-TOF peptide mass fingerprinting (PMF) ✓ Spectral peaks that were specific to the oocyst and sporozoites	[25]
2007	Large-scale global proteomic analysis of non-excyted and excyted *C. parvum* sporozoites	✓ Identification of around 200 proteins, representing about 6% of the predicted proteome ✓ Twenty-six proteins were found to have significantly higher expression levels post-excystation relative to unexcysted oocyst	[27]
2008	In-depth analysis of the expressed protein repertoire of *C. parvum*	✓ A total of 642. 282 and 1154 non-redundant proteins were identified from the 1-DE, 2-DE, and MudPIT analyses, respectively ✓ A total of 1237 non-redundant proteins were identified from excysted oocysts and sporozoites	[28]
2010	Proteome analysis for identifying the key components of the *C. parvum* oocyst wall	✓ COWPs constitute about 75% of the proteins identified in the oocyst walls ✓ COWP1 is the dominant oocyst wall protein	[29]
2013	Proteome analysis of *C. parvum* sporozoites	✓ In total, 135 hits were recorded from the analysis of all 20 gel slices, 41% of which were unique hits for *Cryptosporidium*	[30]
2015	Proteomic analysis of rhoptry-enriched fractions from *C. parvum*	✓ Twenty-two potential novel rhoptry proteins were detected ✓ Novel candidate proteins may be considered targets for researching the invasion pathway of *C. parvum* and the pathogenic mechanisms of rhoptry proteins	[31]
2021	Proteomic analysis of *C. andersoni* oocysts before and after excystation	✓ A total of 1586 proteins were identified ✓ A total of 17 of 1586 were differentially expressed proteins (DEPs) upon excystation and had multiple biological functions associated with control of gene expression at the level of transcription and biosynthetic and metabolic processes	[32]
2021	Proteomic analysis of *Cryptosporidium* spp. from clinical samples	✓ Utility of the purification method for oocysts from clinical stool samples ✓ Implementation of MALDI-TOF MS for clinical sample analysis	[35]
2021	Assessing the effectiveness of cow colostrum for treating cryptosporidiosis in calves and its impact on serum proteomes	✓ The use of colostrum in the treatment of cryptosporidiosis affects the serum proteomes of calves ✓ Serum amyloid A was the most altered proteome in the sera of calves with colostral treatment	[33]
2021	Characterize the changes to the proteome induced by *C. parvum* infection	✓ Among 4406 proteins identified, 121 proteins were identified as differentially abundant in *C. parvum* infected HCT-8 cells compared with uninfected cells ✓ A wide range of functional proteins that participate in host anti-parasite immunity or act as potential targets during infection provides new insights into the molecular mechanism of *C. parvum* infection	[34]
2021	Investigation of the underlying biochemical interaction in C57BL/6J mice infected with *C. parvum*	✓ Glycolysis and glutaminolysis were significantly impacted in the jejunum and ileum during cryptosporidiosis ✓ Gut microbiome response to cryptosporidiosis was detected via increased levels of D-amino acids and SCFAs ✓ Ability of multi-omics to contribute a robust understanding of gut infections and demonstrates the previously unreported infection interactomics as the parasite passes through the gut	[26]

**Table 4 ijms-24-12867-t004:** Comprehensive Overview of Techniques and Resources in *Cryptosporidium* Proteomics Research.

Specimens	Strain	Technique	Search Engines	Reference Genome	Protein Coding Genes in Reference Genome
*C. parvum* and *C.muris* oocysts	Iowa and RN66	MALDI-TOF peptide mass fingerprinting (PMF)	-	-	-
*C. parvum* sporozoites	Iowa	MALDI-TOF MS	-	-	-
*C. parvum* sporozoites non-excysted and excysted	ISSC162	Combination of LC-MS/MS and iTRAQ isobaric labeling	ProQUANT software 1.1 (Applied Biosystems, Foster City, CA)	*C. parvum* Iowa type II	3941
*C. parvum* excysted oocyst/sporozoite	Iowa	Three independent platforms: 1-DE LC-MS/MS, 2-DE LC-MS/MS, and MudPIT	MASCOT search tool, SEQUEST algorithm version 27	*C. parvum* Iowa type II	3941
*C. parvum* sporozoites	Iowa	SDS-PAGE and LC-MS/MS	MASCOT search tool	*C. parvum Iowa type II* *C. hominis TU502*	3941 3886
*C*. *parvum* oocysts	Iowa	LC-MS/MS	SEQUEST search tool, NR database at the NCBI	*C. parvum Iowa type II* *C. hominis TU502*	3941 3886
*C. parvum* Isolate	Iowa	SDS-PAGE and LC-MS/MS	MASCOT in the NCBI, CryptoDB v5.0, and EupathDB v2.16 databases	*C. parvum* Iowa type II	3941
*C. andersoni* oocysts	- ^1^	SDS-PAGE and LC-MS/MS	MaxQuant search engine (v.1.5.2.8), UniProt database	*C. andersoni* 30847	3876
*Cryptosporidium* spp.	-	MALDI-TOF MS	flexControl software on a microflex LT/SH MALDI-TOF (Bruker Daltonik GmbH)	*C. parvum* Iowa type II *C. hominis* TU502	3941 3886
*Cryptosporidium* spp.	-	Label-free proteomic quantification techniques and LC-MS/MS	Maxquant search engine (v.1.5.2.8, Max Planck Institute of Biochemistry, Munich, Germany)	*C. parvum* Iowa type II *C. hominis* TU502	3941 3886
*C. parvum*	-	Multi-omics approach: GC-MS and LC-HR-MS	The Protein Discoverer 2.2 (Thermo Fisher Scientific, Bremen, Germany) and Sequest HT search engines	*C. parvum* Iowa type II *C. hominis* TU502	3941 3886

^1^ Unknown strains from clinical samples.

**Table 5 ijms-24-12867-t005:** Chronological Advancements in Transcriptomic Analysis of *Cryptosporidium* Species.

Year	The Greatest Milestone	Outcome of Study	Reference
2011	Construction and analysis of full-length cDNA library of *C. parvum*	✓ Cluster analysis revealed nine distinct clusters ✓ Expression of transcription-related genes peaked at 2 h, proteins involved in translation dominated at 6 h, while transcripts related to structural genes, such as myosin and tubulin, were most prevalent at 12 h post-infection. An increase in transcript abundance, coinciding with the sexual reproduction of the parasites, was observed from 36 to 72 h ✓ *C. parvum* has a more complex system of transcription regulation than previously thought	[40]
2012	CpArray15K in profiling the gene expressions in the oocysts of *C. parvum* and their responses to UV irradiation	✓ The proteasome and ubiquitin-associated components were highly active, implying that oocysts might employ protein degradation pathways to recycle amino acids in order to overcome the inability to synthesize amino acids de novo ✓ Energy metabolism in oocysts was featured by the highest level of expression of the LDH gene ✓ UV irradiation of oocysts results in increased activities in cytoskeletal rearrangement and intracellular membrane trafficking	[37]
2016	New insights into the intracellular development of *C. parvum*	✓ Identification of genes that are involved in apoptosis, cell growth, and division are differentially expressed in infected cells ✓ Total of 696 genes to be differentially expressed in infected and uninfected cultured cell monolayers ✓ Upregulation of host heat-shock genes and genes for Cysteine X Cysteine (CXC) chemokine	[41]
2017	Discovery of parasite RNA Transcripts delivery to infected epithelial cells during cryptosporidiosis	✓ Transcripts were found to be transported into the nuclei of host epithelial cells during infection ✓ Cdg7_FLc_0990 showed a unique nuclear delivery facilitated by the heat-shock protein 70-mediated nuclear import mechanism ✓ Cdg7_FLc_0990 was overexpressed in intestinal epithelial cells; it led to significant changes in the expression levels of specific genes	[38]
2018	RNA-Seq insights from intestinal proliferating stages to infectious sporozoites	✓ Significant differential gene expression between proliferating stages in the intestine and infectious sporozoites ✓ Identification of 3774 protein-encoding transcripts in *C. parvum* and 173 genes (26 coding for predicted secreted proteins) upregulated in sporozoites	[43]
2019	Comparison of gene expression in the sporozoite and intracellular stages of *C. parvum* by RNA-Seq	✓ Significant differences between the transcriptomes expressed outside and inside the host cell ✓ The oocyst transcriptome is less diverse than in the host cell ✓ Genes significantly overexpressed in oocysts show evidence of specialized functions not found in other Apicomplexa	[42]
2021	Analysis of Long Non-Coding RNA in *C. parvum*	✓ A total of 396 novel lncRNAs were identified, mostly with an antisense character, of which 86% showed differential expression ✓ A positive correlation between lncRNA and upstream mRNA expression was observed	[44]
2021	Transcriptome analysis of the ileocecal tissue infected with *C. parvum* for exploring its potential to induce digestive adenocarcinoma in a rodent model	✓ Downregulation of the expression of defensin, an anti-microbial target of the parasite in response to *C. parvum* infection, was observed in the transformed tissues ✓ Identification of immune suppressor cells and accumulation of pro-inflammatory mediators speculates that chronic inflammation induced by persistent *C. parvum* infection assists in the development of an immunosuppressive tumor microenvironment	[39]
2022	Whole transcriptome analysis of HCT-8 cells infected by *C. parvum*	✓ Identification of 393 dif-lncRNAs, 69 dif-miRNAs, and 115 dif-mRNAs at 3 hpi, and 450 dif-lncRNAs, 129 dif-miRNAs, 117 dif-mRNAs, and 1 dif-circRNA at 12 hpi ✓ dif-mRNAs were significantly enriched in nutritional absorption, metabolic processes, and metabolism-related pathways, while the dif-lncRNAs were mainly involved in the pathway-related and immune-related pathways ✓ dif-miRNAs and dif-circRNA were significantly enriched in apoptosis and apoptosis-related pathways	[45]
2022	Dual transcriptomics to determine IFN-g-independent transcriptomic response to *C. parvum* infection	✓ STAg treatment reduced *C. parvum* Iowa II oocyst shedding in IFN-g-KO mice ✓ Transcripts for type I interferon-responsive genes were more abundant in *C. parvum*-infected mice treated with STAg ✓ *C. parvum* transcript abundance was highest in the ileum, and mucin-like glycoproteins and the GDP-fucose transporter were among the most abundant	[47]
2023	Transcriptomic analysis of infectivity of *C. hominis* and *C. parvum*	✓ *C. parvum*-infected cells showed higher levels of signal using Sporo-Glo™ than *C. hominis*-infected cells ✓ Investigation of the transcriptomic landscape for the *C. hominis* and *C. parvum* assessing the gene expression of 144 host and parasite genes gene expression being at high levels, the levels of putative intracellular *Cryptosporidium* gene expression were low, with no significant difference from controls	[46]

## Data Availability

The data presented in this study are available on request from the corresponding author.

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
