# Peer review of "Investigating Cryptosporidium spp. Using Genomic, Proteomic and Transcriptomic Techniques: Current Progress and Future Directions"

_ijms, 2023, doi:10.3390/ijms241612867_

Round 1
Reviewer 1 Report
Journal
IJMS (ISSN 1422-0067)
Manuscript ID
ijms-2509659
Type
Review
Title
Investigating Cryptosporidium spp. Using Genomic and Proteomic Techniques: Current Progress and Future Directions
General and major comments
The manuscript is scientifically correct; however, I think some changes can be made to improve it and make it more interesting for the scientific community.
It could be interesting to read a future perspectives section at the end of the manuscript, like a conclusion with a short summary of what is done and what the authors think is not- but necessary for a better understanding of Cryptosporidium infection/treatment or diagnostic.
The Ms is complete, but I think it needs to be presented in a more “friendly” reading format; otherwise, it is tedious to read and finish. I suggest adding subtitles to different sections representing event/s the authors consider most relevant will help with this.
What happened with other omics techniques, such as transcriptomics, metabolomics or lipidomic? I think this question is the first thing that comes to my mind when I finish the Ms. I strongly suggest adding more information on this and generating an omics review, maybe? That will be more useful and interesting for the community and better to discuss. However, if the authors disagree, I would like to see a short section mentioning the other omics techniques and how all of them are complemented.
Minor
1.
Lines 46-52: all this data is included in reference 6? If not, please add more references.
Line 53: you are talking about whole-genome sequencing advantages on genomic structure and diagnostic. What do you mean by genomic structure? I suggest adding as advantages:
1. identify genetic variants.
2. can provide valuable information about drug responses.
3. identifying therapeutic targets.
4. can provide insights into population genetics and evolution. It helps identify genetic variations, mutations, and patterns of genetic diversity within and between populations.
5. It helps identify resistance genes, understand the mechanisms of pathogenicity, and track the spread of infectious diseases (epidemiological studies).
Line 59: delete the .
Line 61: Please add references.
Line 61-64: Can you modify this sentence and include the last one in the same sentence? such as: Therefore, comprehending the fundamental mechanisms of drug transport to the parasite and identifying distinct targets are crucial for developing effective therapeutic agents, making proteomic studies invaluable in this pursuit.
2.
2.1. Exploring Cryptosporidium Diversity and Evolutionary History through Whole Ge-69 nome Sequencing
This section is too long. Summarize the most relevant events and generate a friendly timeline with those events you consider most relevant. Table 2 is very complete I think that a simple timeline that only represents the outcome of the discovery in Crypto together with the table could be reasonable to see in the Ms.
Line 75-81: Is all this detail necessary? Please evaluate to write this in a simple way and easy/pleasant read.
3. Proteome section.
I think that this section needs to be separated into different sections like the previous one.
Author Response
Dear Reviewer,
Thank you for all the valuable suggestions, and we truly appreciate the effort you put into reviewing our manuscript. Below, we provide our responses written in bold to each part of your review.
General and major comments
The manuscript is scientifically correct; however, I think some changes can be made to improve it and make it more interesting for the scientific community.
It could be interesting to read a future perspectives section at the end of the manuscript, like a conclusion with a short summary of what is done and what the authors think is not- but necessary for a better understanding of Cryptosporidium infection/treatment or diagnostic.
Our response: Thank you for this suggestion. The "Future perspective" section has been added. Futhermore, the added “Discussion” section also addresses the reviewer's suggestions.
The Ms is complete, but I think it needs to be presented in a more “friendly” reading format; otherwise, it is tedious to read and finish. I suggest adding subtitles to different sections representing event/s the authors consider most relevant will help with this.
Our response: We agree with the reviewer, and that's why we have added subtitles to each section.
What happened with other omics techniques, such as transcriptomics, metabolomics or lipidomic? I think this question is the first thing that comes to my mind when I finish the Ms. I strongly suggest adding more information on this and generating an omics review, maybe? That will be more useful and interesting for the community and better to discuss. However, if the authors disagree, I would like to see a short section mentioning the other omics techniques and how all of them are complemented.
Our response: Thank you for this valuable suggestion. We agree with the reviewer that our manuscript lacks of the other omics studies used in Cryptosporidium fields. Therefore, we have decided to add a section dedicated to transcriptomics.
Minor
Lines 46-52: all this data is included in reference 6? If not, please add more references.
Our response: We include the following reference (L 53):
https://www.scielo.br/j/rimtsp/a/fyQTYWknrhMMDMtNfx4MYtn/?lang=en
Line 53: you are talking about whole-genome sequencing advantages on genomic structure and diagnostic. What do you mean by genomic structure? I suggest adding as advantages:
- identify genetic variants. 2. can provide valuable information about drug responses.
- identifying therapeutic targets. 4. can provide insights into population genetics and evolution. It helps identify genetic variations, mutations, and patterns of genetic diversity within and between populations 5. It helps identify resistance genes, understand the mechanisms of pathogenicity, and track the spread of infectious diseases (epidemiological studies).
Our response: Thank you to the reviewer for bringing attention to the unfortunate phrase we used. Instead of "Genomic structure," we should have used "genomic structure of DNA," which refers to the discrete regions containing genes that encode proteins or RNA. Your valuable feedback has helped us rectify this mistake, and we sincerely appreciate your input. Therefore, we have decided to remove that unfortunate expression to maintain coherence in the text.
Line 54-60: As per the reviewer's suggestion, we have incorporated a few sentences highlighting the advantages of Whole Genome Sequencing (WGS).
Line 59: delete the
Our response: Due to the addition of the "transcriptomics" paragraph in the introduction, we have decided to shorten the section describing proteomics in the introduction to maintain a balanced representation of genomics, proteomics, and transcriptomics.
Line 61: Please add references.
https://doi.org/10.1016/S1368-7646(02)00011-0
Our response: Done in L 69.
Line 61-64: Can you modify this sentence and include the last one in the same sentence? such as: Therefore, comprehending the fundamental mechanisms of drug transport to the parasite and identifying distinct targets are crucial for developing effective therapeutic agents, making proteomic studies invaluable in this pursuit.
Our response: We have made a deliberate decision to modify the entire paragraph so that it better aligns with the overall content and context of the Introduction.
2.1. Exploring Cryptosporidium Diversity and Evolutionary History through Whole Ge-69 nome Sequencing
This section is too long. Summarize the most relevant events and generate a friendly timeline with those events you consider most relevant. Table 2 is very complete I think that a simple timeline that only represents the outcome of the discovery in Crypto together with the table could be reasonable to see in the Ms.
Our response: We agree with reviewer and we shorten the section 2.1. Furthermore, we have prepared a friendly timeline as a figure (Fig.1).
Line 75-81: Is all this detail necessary? Please evaluate to write this in a simple way and easy/pleasant read.
Our response: In response to the previous reviewer's suggestion, we have shortened section 2.1. Therefore, after careful consideration, we have decided that the sentence in its current form is not necessary, and thus, we have removed it.
- Proteome section.
I think that this section needs to be separated into different sections like the previous one.
Our response: Done.
Reviewer 2 Report
The manuscript by DÄ…browska et al. is well written easily understandable and provides a thorough overview about representative studies describing accomplishments in the field of Cryptosporidium research using genomics and proteomics. The tables are very helpful, and the text does a good job for provide a general overview about the results of several investigations in study of the Cryptosporidium spp. genomics and proteomics. Nevertheless, below you have some suggestions in order to improve it:
L23, L349: „we summarize” „we discuss” – I would like to suggests to the authors to avoid the using of personal mode verb formulations, it is not so characteristic for the scientific style;
L25: the using of the “major accomplishments” term in the abstract section is unclear. Please clarify. In addition, the authors must complete this section with comprehensive sentences presenting the main results and further strategies and perspectives in Cryptosporidium genomics and proteomics research (maybe rephrasing the sentences from the 595-599 lines would be useful).
L37: instead of “developing”, please use “low-incoming”
L41: the used reference at the end of the paragraph presenting Cryptosporidium taxonomy (Kotková et al., 2016) is outdated. Regarding the total number of species, genotypes and their zoonotic potential please consult and cite https://doi.org/10.1016/j.ijpara.2021.08.007. (the reviewer is not the author of this article).
L67: according to the reference list the manuscript process the content of 37 articles. I would like to strongly suggest to the authors to provide a short „Materials and methods” section with the description of the data mining process, the eligibility criteria of the cited works, and the and rejection criteria of the manuscript that were not included in the article after database search, number of chosen and rejected works, keywords used in the data mining process and in the different databases. The articles have been selected with a solid and clear methodology? If yes, what?
A paragraph with a complete and clear description of the bibliographic research is mandatory for this valuable a review in order to increase its value! I would like to see a revised version that includes a materials and methods section!
The presented paragraphs are well structured and presented. However, the reader feeling is that, in majority of cases, the provided data is a large list of information presenting the main results of the processed studies, without any critical discussion. So, this aspect can be improved.
L595: please include the mentioned information from this line as Conclusions
L621: please carefully revise and italicize the scientific name of species throughout the reference list (e.g. Cryptosporidium ubiquitum, etc.)
Author Response
Dear Reviewer,
Thank you for all the valuable suggestions, and we truly appreciate the effort you put into reviewing our manuscript. Below, we provide our responses written in bold to each part of your review.
The manuscript by DÄ…browska et al. is well written easily understandable and provides a thorough overview about representative studies describing accomplishments in the field of Cryptosporidium research using genomics and proteomics. The tables are very helpful, and the text does a good job for provide a general overview about the results of several investigations in study of the Cryptosporidium spp. genomics and proteomics. Nevertheless, below you have some suggestions in order to improve it:
L23, L349: „we summarize” „we discuss” – I would like to suggests to the authors to avoid the using of personal mode verb formulations, it is not so characteristic for the scientific style;
Our response: Thank you very much to the reviewer for bringing this mistake to our attention. We have corrected the sentences containing these incorrect expressions to the impersonal mode.
Improved for “This review presents an overview of the significant achievements in Cryptosporidium research by utilizing genomics and proteomics approaches” (L 23-25)
Improved for “In this review, recent advances in methods for isolating and enriching Cryptosporidium genomic DNA from clinical specimens for whole-genome sequencing are discussed” (L 295-297)
L25: the using of the “major accomplishments” term in the abstract section is unclear. Please clarify.
Our response: Thank you for this comment. We mproved this expression for “This review presents an overview of the significant achievements in Cryptosporidium research by utilizing genomics, proteomics and transcriptomics approaches” (L 23)
In addition, the authors must complete this section with comprehensive sentences presenting the main results and further strategies and perspectives in Cryptosporidium genomics and proteomics research (maybe rephrasing the sentences from the 595-599 lines would be useful).
Our response: Thank you for the valuable suggestion. We fully agree with the reviewer's proposal in this matter. After careful consideration, as a result, we have now added comprehensive sentences summarizing the main results and presenting strategies and perspectives for future research on Cryptosporidium genomics, proteomics, and transcriptomics in the final part of the manuscript, namely the Discussion and Future Perspectives sections. The sentences from lines 595-599 have been rephrased and incorporated into the section to enhance its content and relevance.
L37: instead of “developing”, please use “low-incoming”
Our response: Done (L 37).
L41: the used reference at the end of the paragraph presenting Cryptosporidium taxonomy (Kotková et al., 2016) is outdated. Regarding the total number of species, genotypes and their zoonotic potential please consult and cite https://doi.org/10.1016/j.ijpara.2021.08.007. (the reviewer is not the author of this article).
Our response: We agree. Done (L 41)
L67: according to the reference list the manuscript process the content of 37 articles. I would like to strongly suggest to the authors to provide a short „Materials and methods” section with the description of the data mining process, the eligibility criteria of the cited works, and the and rejection criteria of the manuscript that were not included in the article after database search, number of chosen and rejected works, keywords used in the data mining process and in the different databases. The articles have been selected with a solid and clear methodology? If yes, what?
A paragraph with a complete and clear description of the bibliographic research is mandatory for this valuable a review in order to increase its value! I would like to see a revised version that includes a materials and methods section!
Our response: Thank you for your valuable suggestion. Our primary objective was to prepare a literature review, specifically focusing on Cryptosporidium omics, rather than conducting a systematic review with meta-analysis. Our selection of articles was based on subjective assessment to highlight those we deemed as groundbreaking in the field, which is why we referred to them as "milestones."
In response to your feedback, we have decided to include additional sentences in the Introduction, where we will explain our selection process, the purpose of this review, and the intended audience (L76-84).
We want to emphasize that although our approach involved subjective judgment, we have invested considerable effort in compiling this comprehensive review. Our aim is to provide a valuable contribution to the understanding of Cryptosporidium omics for readers interested in this subject.
The presented paragraphs are well structured and presented. However, the reader feeling is that, in majority of cases, the provided data is a large list of information presenting the main results of the processed studies, without any critical discussion. So, this aspect can be improved.
Our response: Thank you for this suggestion. In response to it, we have added headings to each paragraph to better organize and perform all aspects of the research. Additionally, we have included a "Discussion" section where we attempted to highlight and critically assess the most significant achievements. Unfortunately, due to space constraints and the nature of some data presented in the manuscript, which includes announcements of genomes and other non-discussable information, we have opted for this format. Furthermore,
L595: please include the mentioned information from this line as Conclusions
Our response: done
L621: please carefully revise and italicize the scientific name of species throughout the reference list (e.g. Cryptosporidium ubiquitum, etc.)
Our response: done
Round 2
Reviewer 1 Report
Thank you very much. I think the Ms is improved.
I did not find Figure 1. Also, in line 462, please change correct the "ang" Could you please submit the complete version?
Many thanks
Author Response
Dear Reviewer,
Thank you very much for accepting the article. I have attached Figure 1, and I have already corrected "ang" to "and" in Line 322.
Thank you once again for your valuable feedback and for accepting our article.
Best regards,
Joanna DÄ…browska
Jacek Sroka
Tomasz Cencek

Reviewer 2 Report
The authors correctly acknowledged all of the raised concerns. Congratulations!
Author Response
Dear Reviewer,
Thank you very much for accepting our article.
All the best and good luck,
Joanna DÄ…browska
Jacek Sroka
Tomasz Cencek